# Application of Forward Error Correction (FEC) Codes in Wireless Acoustic Emission Structural Health Monitoring on Railway Infrastructures

**Evangelos D. Spyrou [1,2] and Vassilios Kappatos [1,*]**

[1] Hellenic Institute of Transport, Centre for Research and Technology Hellas, 57001 Thermi, Greece; espyrou@certh.gr

[2] Department of Informatics and Telecommunication, University of Ioannina, 45110 Ioannina, Greece

[*] Correspondence: vkappatos@certh.gr

**Abstract:** Structural health monitoring (SHM) has been extensively used in the railway industry, with applications ranging from railway infrastructures to carbody shells. An SHM method that dominates monitoring procedures is Acoustic Emissions (AE). The utilisation of the AE method could use a significantly large amount of data, collected and forwarded to terminal computers using wireless communications. Nowadays, the use of 5G is taking over traditional wireless such as Wi-Fi and 4G telecommunications. However, errors in the transmission due to noisy channels may be experienced. The SHM system may result in the wrong detection of a potential defect in a railway infrastructure with dangerous consequences, such as derailment. Hence, methods for adequately dealing with these errors need to be established, such as Forward Error Correction (FEC) codes. In this paper, we suggest the use of the wireless FEC codes applied to a number of deployed AE devices, in order to perform correction at the transmissions. We investigate the new POLAR codes and compare this method with the Reed-Solomon (RS) codes. We present simulations that the POLAR codes are more efficient with trials using the AFF3CT simulator.

**Keywords:** SHM; railway; wireless; 5G; FEC codes; noisy channel; POLAR; Reed-Solomon

## 1. Introduction

Essentially, structural health monitoring (SHM) is about monitoring infrastructure and providing predictions regarding fatigue life. SHM acts by sending waves from sources ranging from acoustic, thermal, or electromagnetic, among others. These methods comprise different implementations and sensors, and they are, therefore, crucial to be knowledgeable about them [1]. SHM is a major trend as an application in the transport domain, and it includes aircraft state monitoring, high-speed train car-bodies, rail tracks, bridges, etc. [2–6]. Here, SHM in carbodies is considered for potential cracks or disjoint parts.

An excellent method for performing SHM is acoustic emission (AE), among others [7]. Acousto-ultrasonics also utilise AE; however, this method constitutes an inspection method [8,9]. There are a number of applications of AE monitoring in different infrastructure [10]. Importantly, AE can be utilised for real-time damage detection via monitoring. AE is essentially the generation of elastic waves due to the release of energy from a localised source within a material composing a structure. Among the main features of AE one can define are the capability of real-time monitoring, high sensitivity in the sensing, global monitoring, source location, high sensitivity to any mechanism that produces stress waves, and passive nature since the energy from the source is encapsulated [11].

In [12], the authors investigated microseismic/acoustic emission source localisation in order to predict dangerous events on complex structures. The main problem is the location of the errors, which are attributed to irregular structure and pre-measured velocity. A velocity-free- microseismic/AE location method is implemented in order to satisfy the

high accuracy location needs in complex three-dimensional hole-containing structures. Zhang [13] conducted an unaxial multi-stage loading test on siltstone specimens, in order to examine the staged evolutionary features during multi-stage loading based on wave velocity imaging and AE monitoring. The comparison and analysis of the behaviour during the stages of pressure maintenance and loading, during multistage loading, they discovered the localisation effect during rock failure. Moreover, as read in [14], in situ monitoring for tasks of Additive Manufacturing (AM) can be undertaken using AE methods.

SHM using AE could comprise monitoring the entire state of a structure using an array of sensors and assessing their parameters. Moreover, monitoring of a single location in a structure can be conducted, such as the growth of a crack in a specific part of the structure, using one or a plethora of sensors. The acquisition of data from AE sensors is of utmost importance since the data is continuous and the data need to be transferred to a dedicated server. Wireless communications have emerged in the field of the SHM and AE in order to transmit and forward the data to a terminal computer. In particular, wireless sensor networks (WSN)s [15–17] and their successors, namely the Internet of Things (IoT) [18], have been assigned a vital role in the next generation of SHM. Wireless technologies such as zigbee [19], Wi-Fi [20] and telecommunication solutions [21] transform SHM into a wireless networked process, where enhanced autonomicity is accomplished. Note that, especially for the wireless connectivity, solutions such as the Micro-SHM [22] offer 4G capabilities as well as Wi-Fi. In zigbee and Wi-Fi communications, it may experience interference due to lossy links. SHM with acoustic emissions usually requires a Data Acquisition Board to perform calculations of the continuous signal that comes from the AE sensors. Thereafter, the signal that exceeds a specific threshold may be sent to a terminal computer for analysis of specific features. The use of a field-programmable gate array (FPGA) often performs this task. The transmission of the packets using wireless technologies needs to be optimised, in order to meet the high data rate of the AE. Hence, the thresholding of the values. 4G and 5G offer high data rates and low latency; thus, they make a good candidate for SHM.

5G, 5th generation mobile network, is a new global wireless standard coming immediately after the 4G networks. In the 5G era, a large volume of data will be exchanged with low latency at a low energy cost. One key concept is to get low residual error rates with fast, flexible, and low complexity decoders. 5G enables a new kind of network that is destined to connect virtually a plethora of devices, objects, and machines. 5G wireless technology is here to accomplish higher multi-Gbps peak data speeds, ultra-low latency, more reliability, extremely large network capacity, increased availability, and a more uniform user experience to more users. The higher performance, which is promised to achieve, in conjunction with the improved effectiveness ensures new user experiences and connects new industries, the railway industry in our case. Nowadays, in the railway sector, there is the LTE-R, which is the Long-Term Evolution (LTE) for railways, which comes as a candidate against Wi-Fi connectivity. With 5G, we aim to surpass the LTE-R and accomplish even better numbers in terms of latency and connectivity, as well as throughput. In the case of the collection of data from AE sensors in monitoring infrastructure, high coverage (connectivity) and low latency is essential. Those features refer to the ultra Reliable Low Latency context. It seems interesting to evaluate the capability of the 5G network to perform to changing condition; however, the high connectivity and low latency with the increased performance make 5G suitable for SHM.

An excellent method for ensuring that the message is transmitted to the basestation is network coding [23]. The simplest case is where relay nodes are utilised that transmit a linear combination of two messages, i.e., the sum. Then, the decoder essentially receives two messages, since it has one message and a combination of the two messages transmitted. Since data is transmitted in such a way, throughput is maximised. Forward Error Correction codes (FEC) is a method in which a data sender adds an Error-Correcting Code (ECC) to the original data for allowing the receiver to detect and correct a limited number of errors in a noisy channel. The FEC consists of two types, the block code, and the convolutional code. Some of the most recognised block codes include the Hamming code, Golay code,

Bose–Chadhuri–Hocquenghem code and Reed–Solomon (RS) code, as we can see in [24] and references therein. These codes are quite simple and have strong features with respect to burst error. Since the assumption is for a 5G network to be deployed, a large amount of data with a short delay and we may have burst errors, block codes are mainly used. The convolutional codes offer less mathematical complexity than block codes and it offers good features in the additive Gaussian noise environment. A typical example is the turbo code [25]. The linear block codes, names POLAR codes are of great importance and proposed recently [26]. Some related work in other domains includes research that is given below.

Domanovitz et al. [27], extended the upper bound of streaming codes to multi-hop relay networks, and further, the described the state-dependent scheme for the L relay scenario and provided the way it accomplishes the upper bound while requiring an overhead in the packet header. Moreover, they showed that the header size is not dependent on the field size utilised by the code. Hence, the gap from the upper bound decreases as the field size increases.

Facenda et al. [28] investigate a network of two source nodes, one relay node and a destination node, whereby the source nodes' objective is the transmission of a sequence of messages, via the relay, to the destination. The latter needs to decode the messages with a strict delay constraint T. The paper begins with the introduction of two significant tools namely, the delay spectrum and the concatenation. The former generalises delay-constrained point-to-point transmission, while the latter permits the combinations of different codes in order to achieve a wanted level of operability. The encapsulation of these tools results in the successful generalisation of the schemes suggested in [29], which paved the way for a novel scheme that allows rate optimality under well-given conditions. This scheme is enhanced with optimisation, in order to improve the accomplished rates in the cases that the conditions for optimality are not satisfied.

In [30], the authors examine the reliable transmissions mechanisms of the QUIC. In particular, they go through the process from the design to evaluation of FEC extensions to QUIC. These schemes are specially designed for scenarios with high delays and lossy links. The design comprises a generalised FEC frame and their implementation details suggest that there is support for the XOR Reed–Solomon and Convolutional RLC error-correcting codes. The separation of the packets, the received packets from the packets recovered by the FEC codes allows the avoidance of the delay in the loss congestion signal. The evaluation showed the application of an experimental design encapsulating the use of different delays and packet loss conditions. The authors show that their design succeeds in the adaptation of the protocol to the network conditions. As it can be derived from the results when the network exhibits a small loss rate and delay FEC acts as an overhead for the download completion. On the other hand, a high loss rate and longer delays allow the FEC to reduce the downloading time by avoiding costly retransmission timeouts. These results show the need for an adaptive FEC with respect to network conditions.

In [31], the authors address the problem of the application of Forward Erasure Correction (FZC) codes to MPEG-4 video sequences sent and received by hosts of a network comprising a satellite link and wireless LAN with 802.11b devices. The scenario includes the video streaming application, which is executed on one end of the satellite link, while a wireless ad hoc network received the multicast video stream at the other end. The concept of this work is the demonstration of the enhancement of the Quality of Service (QoS) when the video is transmitted in the network. The metrics that were being measured were the packets loss, the reception delay, and the bandwidth occupancy overhead imposed by the use of the FZC.

In this paper, two FEC schemes are employed, in order to apply them for AE in carbodies. We assume that the communication of these networked devices is 5G, which could be the next generation of an already existing wireless AE device, such as the Micro-SHM system that uses 4G. Thereafter, the two schemes are compared, namely the Reed–Solomon (RS) and the POLAR codes for such an application. The emergence of the 5G

communication with these codes can provide more reliable links even in the presence of noise or interference. More specifically, simulations of throughput, Bit-Error Rate (BER) and Frame-Error-Rate (FER) comparison between the two schemes are performed, in order to indicate the one that is more effective in our application.

The contributions of this paper are the following:

- An application of FEC codes has been described, which aims at the transmission of the features of an SHM device without errors.
- After the assumption that 5G communication is used in an SHM application, two FEC schemes have been compared, one being new and the other efficient, namely POLAR and RS codes, respectively.
- The POLAR codes were shown as the best candidate due to metrics that have been used such as the FER, BER, and throughput.

The remainder of this paper is as follows: Section 2 gives the AE setup scenario, Section 3 gives the carbody shell application, Section 4 provides a brief description of the FEC codes, Section 5 gives the results of our simulations and Section 6 gives the conclusions and future work.

## 2. AE SHM Setup

In this section, the scenario of acquiring values from the AE recordings and transmitting them through the wireless interface of the AE module are provided to the reader. Initially, an explanation of the AE module is given, and later the wireless setup scenario is given. The assumption is that we have a number of AE modules, which communicate in a wireless manner, in order to send their signal to a terminal computer we call the sink. This assumption is not pivotal, and we proceed with it to leverage our application.

### 2.1. AE Module

The Micro-SHM [22] is an AE system, which is cost-effective and reliable, targeting SHM applications. It is an ideal candidate for the purpose of our application since it can remotely monitor the joints of the carbody and the shell. Moreover, it can be used to monitor specific infrastructures such as railway bridges and tracks. This remote system is a tool, which offers data management and analysis procedures for AE data recording. The Micro-SHM has wireless capabilities to support remote monitoring and it also includes two channels for AE. In the configuration we are using for this paper, a 2MSPS sampling was used with 2k samples depth. An analogue filter was used from 10 kHz to 1 MHz. Additionally, a digital butterworth bandpass filter was used, from 80 kHz to 200 kHz. This was to avoid any external source of noise that may take place when the monitoring procedure is active.

There exist different options when it comes to wireless communication. Using the Wi-Fi or 4G capabilities that the system offers, we can transmit a packet with the values coming from the raw signal. Essentially, we may have to break the produced signal into a number of packets, and transmit them in order, so that the basestation is able to reconstruct the signal from the packets. The file that is produced by the Micro-SHM consists of the time relative to the first hit and the signal amplitude. The AE signal is continuously produced since it has to be able to detect cracks in real-time. Hence, we are considering a continuous approach for the transmission of the signal values.

On the other hand, the Micro-SHM and its accompanying software can obtain a number of AE features, which correspond to the measurements that have been taken in real-time from the sensors. The acoustic emission defect sources can be simulated using pencil lead breaks to composite materials and the sensors will output the generated signal as it appears in Figure 1, where the reader can see the response from an AE sensor with and without a defect, respectively.

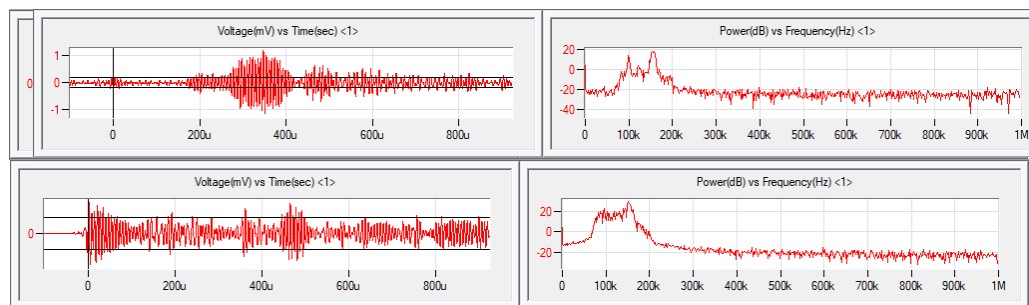

**Figure 1.** Response of AE sensor with (**up**) and without (**bottom**) defect.

Here, we may assume that these measurements are transformed to values at the device level. The most important of these features can be found in the manual of the Micro-SHM and AEWin software that accompanies it. Note that these parameters are sent to the terminal computer in the form of minimum, maximum and average values of the AE hits. Hence, we can put them in a struct with arrays of three values per feature to form a packet.

The nature of the AE monitoring requires that the features are promptly sent to the control centre in the case of a defect, especially if a serious one exists. Subsequently, the AE system that monitors the carbody needs to be equipped with the communication means to send the features without errors. This means that mechanisms, which ensure that the packets are sent without being erroneous, should be available and operate during the transmission operation. In the next parts of the paper, we provide the communication of the AE system, as well as the method to overcome errors in the communication.

### 2.2. Communication

Here, the assumption is that the 5G connectivity (instead of 4G) to the railway infrastructure including carbody and the AE devices as well as the passengers can connect to the same network for data transmission. This assumption is a natural progression of the AE systems in terms of telecommunications. Hence, there might be packet loss from the massive connections that will be established. We can see the configuration of the AE devices in Figure 2, whereby each device communicates with each other via a 5G infrastructure.

Here, the simplest case is taken onboard, meaning that direct device-to-device communication is not considered. Instead, the 5G infrastructure is used to transmit the packet, which can be also used for other transmissions as well. Hence, bursty traffic may be experienced. With the FEC codes, one encoder and one decoder for the transmission of data are considered. The encoder is one 5G device and the decoder can be another 5G device. Thus the data will be passed to another Micro-SHM device in order to perform some kind of computation, which could be federated learning [32].

The packet of the communication, except the header, will include the payload as well, which will consist of a structure of arrays, which will individually represent an AE feature with the three aforementioned values, namely the maximum, minimum, and average values. The communication will be undertaken in such a way, in order for the maximum size of a packet will be serviced.

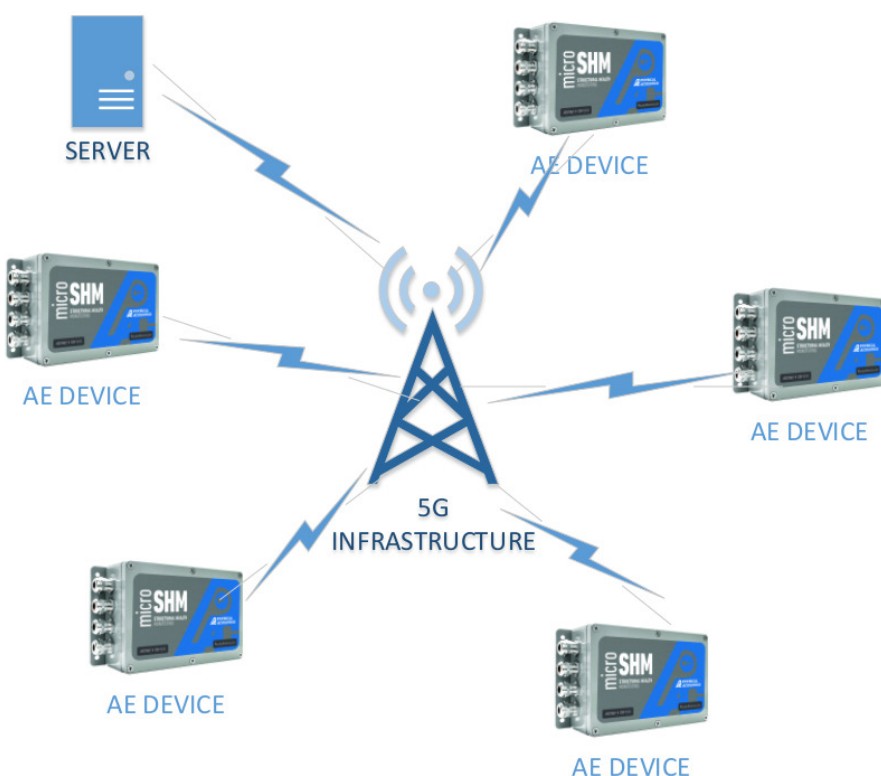

**Figure 2.** Telecommunication configuration.

## 3. Carbody Shell Application

Nowadays, the transportation domain has been substituting metal parts in carbodies, giving way to the use of parts made of composite materials [33]. Composites constitute the new trend in the transport industry because of their primary feature, which comprises strength, stiffness, and reduced weight [34]. These aforementioned features render composite materials a high preference competitor for the railway industry [35].

Railway transportation is essentially one of the cleanest transportation means. However, further enhancements and changes can be made. The environmental footprint of the railway transportation is one aspect that needs to be optimised. Another aspect is the impact of the energy spent by the carbodies' use phase due to their weight [36]. Pruning of the rail carriage mass, which currently consists of metal components, is the primary change for the next generation of trains, in order to accomplish energy savings. Weight reduction could take place by substituting the steel with composite parts [37].

Here, we briefly present our work [35], which was the SHM of composite materials, which would be utilised in rail carbodies. Essentially, this work was a prime investigation of the AE and accelerometers methods to continually assess carbodies. The point was to examine these methodologies for permanent installation on the carbodies and their impact on locating problems and faults.

## 4. Selection of FEC Codes for the Railway Application

FEC describes a methodology whereby the sender of some data includes an ECC to the initial data, in order to permit the receiver to detect and correct a number of errors that appear in a noisy channel. The procedure is given in Figure 3. This procedure can be applied on occasions when there are multiple receivers. Here, we have a noisy channel whereby messages are difficult to be sent. We experience a drawback of the packet increase with the addition of an ECC; however, the ECC gives the advantage that the receiver will be able to correct certain errors that are caused by noise or interference. This gives the

result that systems, which include ECC can suffer some extent of a bit-error-rate (BER) in a given signal-to-noise ratio (SNR).

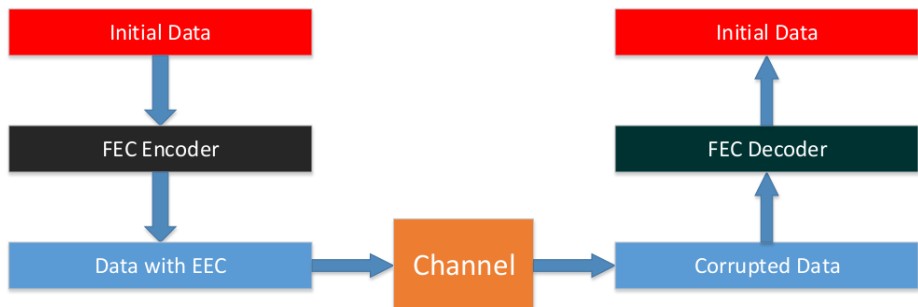

**Figure 3.** FEC scheme.

As we have seen earlier, the FEC is divided into two types: block code and convolutional code. Here, we will deal with the Reed–Solomon (RS) and POLAR schemes. The RS scheme is proposed since it is the most typical and efficient scheme among the FEC schemes. Moreover, we use the POLAR codes, since they are relatively new and adequate for 5G communication. Other codes that could be utilised are the turbo and low-density parity-check (LDPC) codes. The turbo codes are used in 3G/4G mobile communications, and they are the first practical codes that reach the maximum channel capacity, which is the theoretical maximum for the code rate that reliable communication is accomplished under the presence of noise. The LDPC is the competitor of the POLAR codes, which are both suggested in 5G communication [38,39]. The point of our paper is to apply it to an AE telecommunication network and not to invent a new FEC scheme.

For information regarding the RS scheme, the reader can see the [40] paper. For a thorough description of POLAR codes and LDPC, the reader is directed to the [26] and [41] papers, respectively. For a brief comparison between POLAR and LDPC codes, the LDPC codes have been in the research domain for a large period of time, while the POLAR codes were introduced in 2009. The LDPC codes encapsulate an iterative decoding process, with which every consecutive attempt at decoding the block of information informs the next, until convergence to a permissible codeword. On the other hand, POLAR codes choose the recovered information block from a list, which is retrieved from the associated parallel successive cancellation decoding processes. In this way, decoding each consecutive bit of information informs the decoding of the next. For both schemes, the decoder of the channel exhibits a higher complexity to a large extent than the encoder, since it uses iterative or parallel decoding processes, which are based on probabilistic representations of the bits that are encoded. This occurs to surpass the uncertainty, which is attributed to noise, interference, and fading [42]. In this paper, POLAR codes are the main focus, which in conjunction with RS codes are checked for efficiency. The POLAR codes description is given below.

### 4.1. POLAR Codes

As we read in [43], the channel polarization transform is used to create POLAR codes. The notion is that by merging and splitting channels at infinite lengths, the channels (bits' positions) will polarize, with some being highly dependable and others being unstable. The channel capacity can be obtained if the information bits are exclusively put into the reliable channels and foreknown bits (typically zeros) are put into the unreliable channels. The goal of POLAR code creation is to identify the Frozen Set, which is a collection of the least trustworthy channels. Multiple building techniques exist, each with a different level of complexity, and due to the non-universal behaviour of POLAR codes, these algorithms require a parameter, namely Design-SNR. However, universal constructions exist.

### 4.1.1. Encoding

The encoder is essentially the polarization transform that the kernel provides, as

$$\mathbf{F} = \begin{bmatrix} 1 & 0 \\ 1 & 1 \end{bmatrix} \tag{1}$$

The Kronecker product of this kernel with itself yields the transform for a larger input size, resulting in POLAR codes with lengths that are powers of two. For a code of length $N$ and $n = \log 2(N)$, the encoder is

$$\mathbf{G} = \mathbf{F}^{\oplus n}, \tag{2}$$

where $\mathbf{F}^{\oplus n}$ is the Kronecker product of $\mathbf{F}$ $n$ times with itself. Thereafter, the encoding occurs in a similar way such as

$$\mathbf{c} = \mathbf{u}\mathbf{G}, \tag{3}$$

with $\mathbf{c}$ being the output codeword, $\mathbf{u}$ being the input block, and $\mathbf{G}$ being the generator matrix.

### 4.1.2. Decoding

Although belief propagation can be used to decode POLAR codes, the usual decoding approach is successive cancellation (SC). The SC decoder may be found immediately from the encoder, where the probabilistic nodes $f$ and $g$ represent the XOR and connection nodes, respectively. The $f$ and $g$ nodes in the LLR domain conduct the following calculations for input Log-Likelihood Ratios $\alpha$ and $b$

$$f(\alpha, b) = \log\left(\frac{e^{\alpha+b} + 1}{e^{\alpha} + e^{b}}\right) \tag{4}$$

$$g(\alpha, b, s) = (-1)^{s}\alpha + b \tag{5}$$

with $s$ being the partial sum, namely the sum of the prior bits that have been decoded and they are taking part in the current $g$ node. The $f$ node can be susceptible to an application of approximation. The LLRs are propagated from right to left as we can see in [43]. By passing the LLRs through the proper $f$ nodes, the first bit, $u_1$, can be directly decoded. After it has been decoded, the $u_2$ can be decoded using a $g$ node, which requires the associated partial sum. Because just $u_1$ is involved, the partial sum is equal to $u_1$. If $u_1$ is frozen, the decoder already knows what it is and can set it to zero. The channel LLRs and $u_1$ are employed to decode $u_2$, resulting in increased $u_2$ decoding reliability. The decoding procedure will continue until all of the nodes have been processed.

An improvement of the performance can be undertaken by using a List-SC decoder. That is, for every decoded bit, the two cases of the decoding to be 0 or 1 are taken. This is accomplished by separating the current decoding path into two other paths, which represent one of the cases. The number of cases across the decoding tree is bounded by the size of the list. After the decoding process is over, the path with the smallest path metric is chosen. More improvement strategies can be encapsulated by the employment of a Cyclic Redudancy Check (CRC) on the paths that remain, with the one satisfying it, to be the right one.

### 5. Simulation—Results

Initially, we add the nomenclature table of the parameters used, which can be found in Table 1.

For our simulations, the AFF3CT toolbox [44] is used, which is a toolbox dedicated to FEC or channel coding. It is written in C++ and it supports a large range of codes including the new POLAR codes. Here, we will compare the RS codes with the POLAR codes, in order to show metrics such as the BER, Frame-Error-Rate (FER), and throughput. The

AFF3CT comes with a simulator that is used for communication chains and targets channel coding levels. It is based on a Monte Carlo method, whereby the transmitter emits frames with random noise by the channel and the receiver attempts to decode the noised frames. The transmitter keeps on emitting frames until it reaches a number of errors that is fixed. A frame error takes place when the initial frame from the transmitter is different from the decoded frame of the receiver. This results in the number of simulated frames increasing, in conjunction with the simulation time, when the Signal to Noise Ratio (SNR) decreases.

**Table 1.** Nomenclature.

| Symbol | Explanation |
| --- | --- |
| -C | input the type of FEC codes we will use |
| -K | number of information bits |
| -N | frame size |
| -m | minimal noise energfy value (dB) |
| -M | maximam noise energy value |
| -s | step between SNR value |
| –chan-type | Additive White Gaussian Noise value |

The AFF3CT simulator works from the command prompt and can take a variety of arguments. Here, we selected a number of the available ones, in order to proceed with an initial simulation of the codes. We used the -C argument, which takes as input the type of FEC codes we will use; in our case RS and POLAR. The argument -K is the number of information bits we set to 1723 and -N is the frame size (bits transmitted over the channel) we set to 2047. This gives us the bit rate as well, which is the division between 1723 and 2047, which gives us the value of 0.8417. The AFF3CT simulator estimates the BER and the FER for an SNR range. The argument -m is the minimal noise energy value (0 dB) to simulate and -M is the maximal noise energy value (4.5 dB). The argument -s is the step between each SNR value we set to 0.5 dB. We also use the –chan-type with the value Additive white Gaussian noise (AWGN). Finally, we use the Phase Shift Keying (PSK) modulation at the modem level. For the simulation, we utilised the standard BER/FER standard [45], which we can see in Figure 4. The complete set of arguments that have been used in this simulation can be seen in Figure 5.

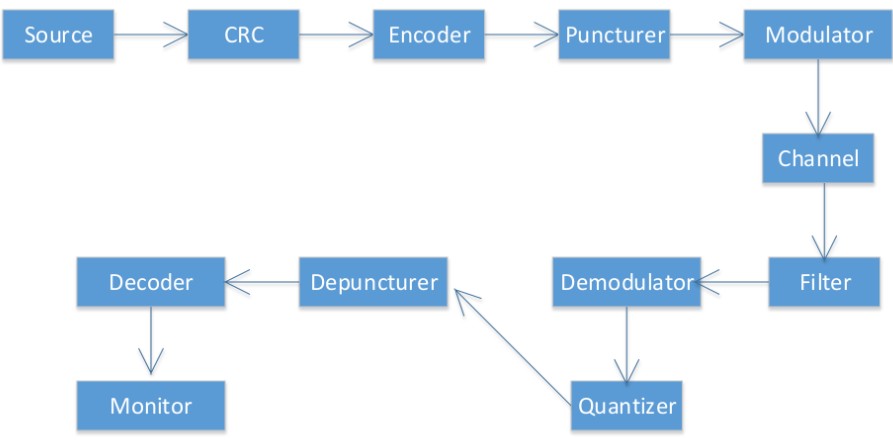

**Figure 4.** Standard BER/FER chain.

Thereafter, a simulation of the RS vs. the POLAR codes was performed, and the results were determined with respect to the BER, FER, and throughput that have been exhibited. The reader can see the results in Figures 6–8. In terms of the BER (Figure 6), we can see a decrease in both RS and POLAR codes as the SNR increases. However, the BER of the

RS codes shows a smoother decrease starting close to 0.7 and ending o 0.147. On the other hand, the POLAR codes commence close to 0.9 BER when the SNR is 1.0 dB, and it ends up to 0 when the SNR is 4.5 dB. In terms of the FER Figure 7, the RS codes exhibit a stationary value ot 1.0 FER throughout the simulation experiment. On the contrary, POLAR codes show a rapid decrease from 1.0 FER to 0.0 FER when the SNR is 4.5 dB. Lastly, as for the throughput, the RS codes show a stable value close to 5 MB/s, while the POLAR codes exhibit a rapid increase from below 4 MB/s at the start to over 10 MB/s at the last value of the SNR.

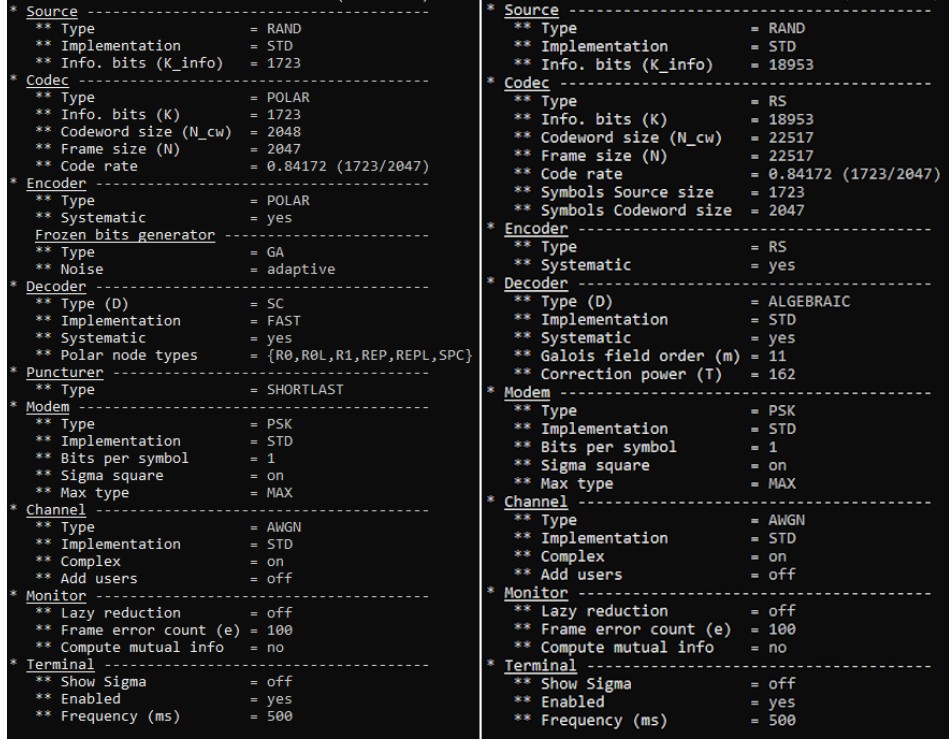

**Figure 5.** POLAR and RS simulation arguments.

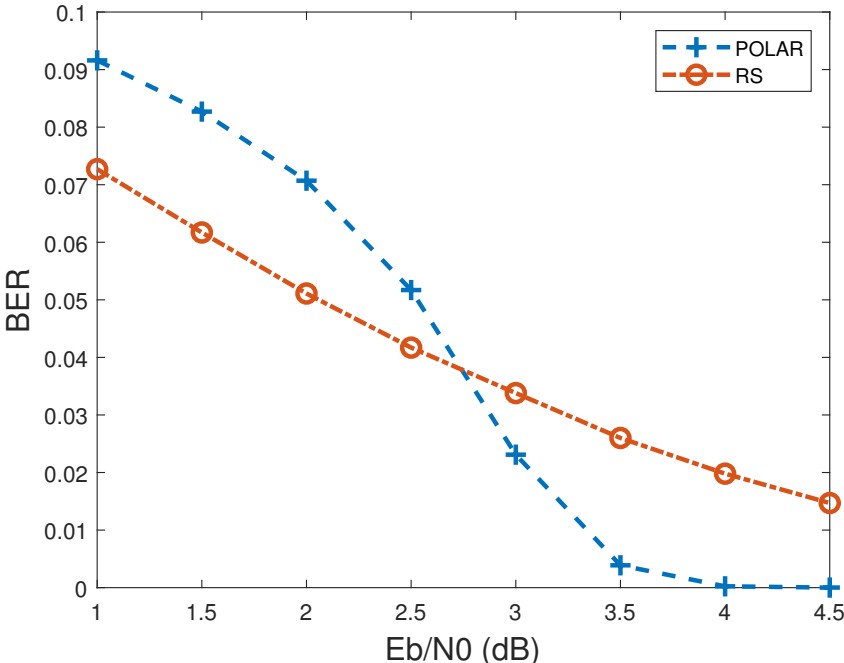

**Figure 6.** BER of different SNR values for RS and Polar codes.

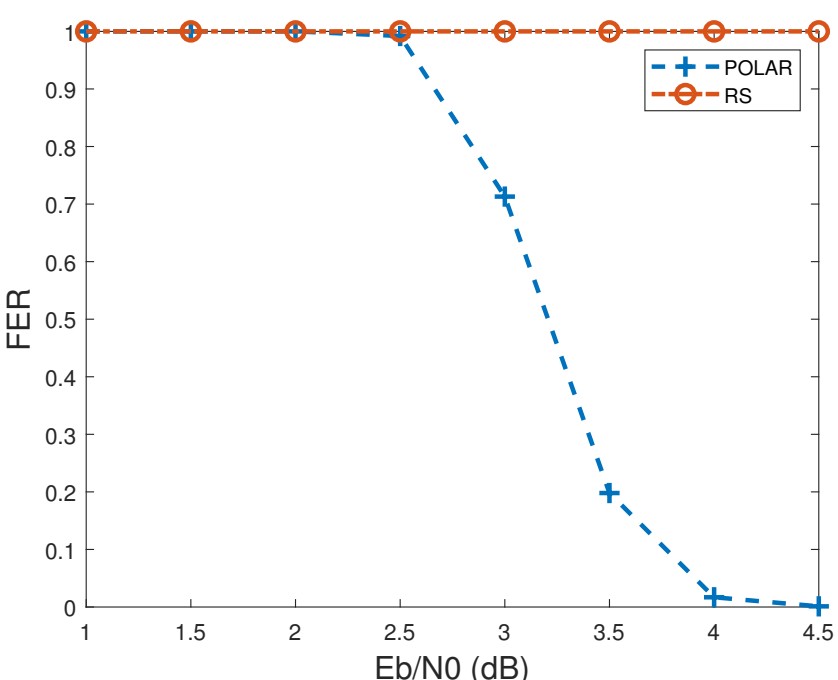

**Figure 7.** FER of different SNR values for RS and Polar codes.

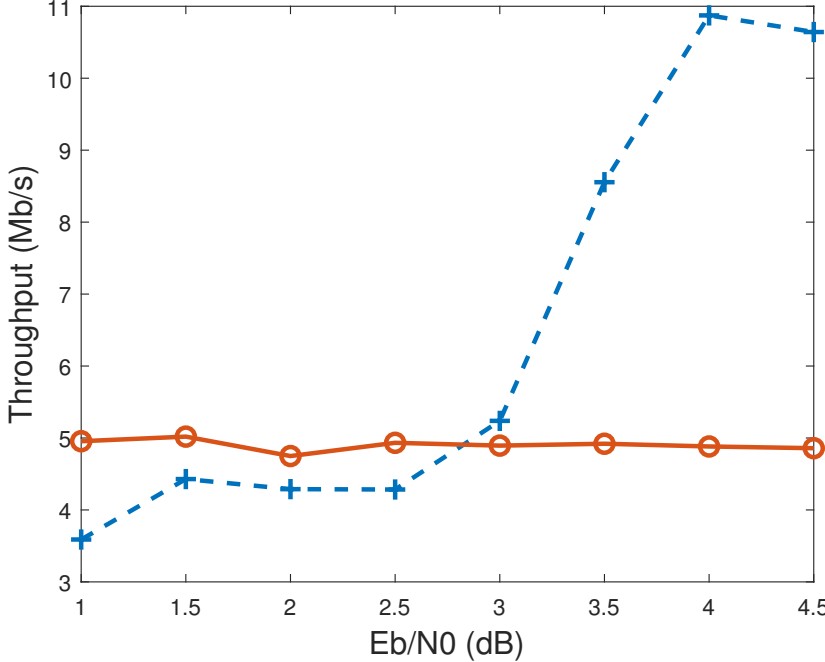

**Figure 8.** Throughput of different SNR values for RS and Polar codes.

Overall, the reader can see that the POLAR codes produce smaller BER and FER with the increase of the SNR value in simulation and higher throughput, which essentially makes them better for the application we are interested in.

## 6. Conclusions and Future Work

In this paper, the use of FEC codes in a railway application of carbody SHM was investigated, which can be extended to other infrastructure of the railway industry as well. A wireless acoustic emission system was described, namely the Micro-SHM system, which obtains specific features and transmits them via telecommunications, in order to

be forwarded in a central processing unit. The assumption is that the transformation of the signal to features occurs in the device. The case is to utilise 5G communication for our communication; note that the Micro-SHM can come with 4G capability.

Due to noisy channels, FEC codes were used to reduce errors in communication. The new Polar codes were encapsulated and were compared with the RS codes, which are good in terms of performance. We encapsulated the AFF3CT simulator to run simulations and we obtained the BER, FER, and throughput for a range of SNR. The POLAR codes are better for the particular application in terms of these metrics.

For future work, the aim is to put the $\pi/2$BPSK modulation to the simulator and run more experiments and comparisons with FEC codes including Turbo and LDPC codes. Moreover, we will also use machine learning to optimise the decoding process of the FEC codes.

Finally, The successive cancellation (SC) decoder used in POLAR codes shows advantages in complexity, but is limited by its bit-by-bit decoding strategy. We aim to investigate a more thorough decoder to overcome the limitations of the current approach. The network construction cost is going to be overcome by using a custom 5G network, which will be utilized for the purpose of the application only.

**Author Contributions:** Methodology, validation, investigation, writing—original draft preparation, data creation: E.D.S. Acoustic Emission Structural Health Monitoring methodology, review, editing, supervision: V.K. All authors have read and agreed to the published version of the manuscript.

**Funding:** This research was co-financed by the European Regional Development and Greek national funds Fund under the framework of the Operational Program «Central Macedonia 2014 2020» (Project code: KMP6-0077768).

**Data Availability Statement:** The data that has been used in this paper were produced from the AFF3CT simulator. The link of this simulator can be found in the references and the manner with which to replicate the data is found in the manuscript.

**Acknowledgments:** 1. This publication has been produced within the project CARBODIN (Car Body Shells, Doors and Interiors). This project has received funding from the Shift2Rail Joint Undertaking (JU) under grant agreement No. 881814. 2. This research was co-financed by the European Regional Development and Greek national funds Fund under the framework of the Operational Program «Central Macedonia 2014 2020» (Project code: KMP6-0077768).

**Conflicts of Interest:** The authors declare no conflict of interest.

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
