# Peer review of "Application of Forward Error Correction (FEC) Codes in Wireless Acoustic Emission Structural Health Monitoring on Railway Infrastructures"

_infrastructures, doi:10.3390/infrastructures7030041_

Round 1

Reviewer 1 Report

Dear Authors

The manuscript is somehow interesting and it may possess sufficient material to be considered as a potential for publication in this journal. The language must be reviewed. The paper organization must be improved. There are major comments that must be addressed by the authors in the revised version.

  1. Please avoid using the active tense starting with “we”, or “our”. It is better to use the passive tense, it is more academic.
  2. The paper lacks a table of nomenclature. Please include it in the revised version.
  3. Please rewrite the conclusions. In fact, reorganise it in the way that shorter and more comprehensive.
  4. What is the novelty of this work? Although I can feel it along the text and results, it is better to clearly mention the novelty of the work. Its contribution to the state of the art is also missing. After reviewing the literature, you must indicate the contribution of the proposed methodology. Please add it in the revised version.
  5. Please merge section 1 and section 2. Basically both describe the literature review.
  6. This paper seriously lacks the demonstrative model. It is all text. When readers looking into this paper, they will be lost for sure. I do recommend the authors to add some perspective of the model.
  7. Methodology must be better explained. In the way that other students and researcher can repeat you work in their own demands.
  8. Lack of examples is serious in your paper. The readers must follow what you have developed.
  9. Still confused about the SHM application in your work. May be more explanations must be presented.

Very Best

The Reviewer

Author Response

Thank you so much for your feedback.

Please find attached our replies.

Reviewer 2 Report

This study used the wireless FEC codes applied to a number of deployed AE devices, in order to perform correction at the transmissions. It is a carefully done study and the findings are of considerable interest. However, I thought it still has some deficiencies and I recommend to a major revision before acceptable publication. Detailed comments are listed below:

--Section 1: A more detailed description of previous research in introduction is needed, especially the published literatures on a number of applications of AE monitoring in different infrastructure, for example: Velocity-Free MS/AE Source Location Method for Three-Dimensional Hole-Containing Structures. Engineering, 2020, 6:827-834. Applying neural-network-based machine learning to additive manufacturing: current applications, challenges, and future perspectives. Engineering, 2019, 5(4): 721-729. Fracture evolution and localization effect of damage in rock based on wave velocity imaging technology. Journal of Central South University, 2021, 28(9): 2752-2769.

--Section 1: The English abbreviation that appears for the first time in the text, please mark its full name

--Section 1: As stated in the text, Wireless communications have emerged in the field of the SHM and AE in order to transmit and forward the data to terminal computer. Has wireless communication technology been used in structural health monitoring?

--Section 1: A variety of wireless communication technologies are listed in the paper, but their role in structural health monitoring is not presented by the author. Are these wireless communication technologies suitable for structural health testing? Please make it clearly.

--Section 1: For the introduction of 5G, the author occupies a lot of space, but its advantages in structural health monitoring are not highlighted.

--Section 2: The author introduced the previous research results, but such description took up too much space. In the literature review part, the author introduced the current research status of the geoscientific literature, the literature review should be concise, try to draw general conclusions through abstracting and summarizing.

--Section 2: The serial numbers of the references should appear in order (e.g. Lines105-Lines116, with reference [36] followed by reference [43]. Lines 77-78, with reference [22] followed by reference [38]), please adjust it.

--Section 3: How are active acoustic emission sources implemented? Are the signal input and output the same? Are there actual pictures to show?

--Section 3: 5G needs to speed up the transmission and response of the network. Therefore, low-frequency signals cannot be used in construction, but the penetration of high-frequency signals is poor. In areas with dense buildings, more base stations need to be built to ensure signal penetration and signal transmission quality. How to overcome this drawback?

--Section 4: As stated in the text, Composites constitute the new trend in the transport industry because of their primary feature, which comprises strength, stiffness and reduced weight. If the use of such composites is not widespread (or is not widespread), How to ensure the rationality of the experimental results in this paper?

--Section 5: FEC techniques tend to reduce the effectiveness of communication, how did the authors overcome this problem?

--Section 6: In future applications, how to reduce coding redundancy degrees and network construction costs? Please make it clearly.

Author Response

(The authors gave the same response as above.)

Round 2

Reviewer 1 Report

Dear Authors

Thanks for your effort to revise the paper based on my comments.

It would be accepted in this format.

Just a small note, please put the table of nomenclature in the beginning of the paper, can be introduction section for instance.

Very best

The Reviewer

Reviewer 2 Report

Revised per comments